# Weaning period and growth patterns of captive Sunda pangolin (*Manis javanica*) cubs

**Dingyu Yan**[1]*, **Xiangyan Zeng**[1], **Miaomiao Jia**[1], **Xiaobing Guo**[1], **Tengcheng Que**[2], **Li Tao**[3], **Mingzhe Li**[4], **Baocai Li**[1], **Jinyan Chen**[1], **Shanghua Xu**[1], **Yan Hua**[5], **Shibao Wu**[6], **Peng Zeng**[2], **Shousheng Li**[2], **Yongjie Wei**[2]

**1** Guangxi Forestry Research Institute, Nanning, Guangxi, P.R. China, **2** Guangxi Terrestrial Wildlife Rescue Research and Epidemic Disease Monitoring Centre, Nanning, Guangxi, P.R., China, **3** Guangxi Institute of Veterinary Research, Nanning, Guangxi, P.R. China, **4** China Wildlife Conservation Association, Beijing, P.R. China, **5** Guangdong Academy of Forestry, Guangzhou, Guangdong, P.R., China, **6** School of Life Science, South China Normal University, Guangzhou, Guangdong, P.R. China

* Yandy6@126.com

**Data Availability Statement:** All relevant data are within the manuscript and its Supporting Information files.

**Funding:** This study was funded by the Nature Science Foundation of Guangxi (2018GXNSFAA294066), the State Forestry

## Abstract

This study tracked and recorded the weight changes of 13 captive Sunda pangolin cubs from lactation to maturity to explored the appropriate weaning time and reveal the rules of its weight growth. SPSS 25.0 was used to build a cubic equation model to fit the body weight change rules of 4 individuals who nonvoluntarily ingested artificial feed (NIAF) at 127 days after birth and 5 individuals who voluntarily ingested artificial feed (VIAF) at 86–108 days after birth. The body weight of NIAF cubs aged 0–120 days and VIAF cubs aged 0–150 days were estimated according to the fitting model. An independent sample T-test was performed on the mean body weight of the two groups during the late lactation period. The results showed that at 105 days after birth, the body weight of the VIAF group was significantly higher than that of the NIAF group (P = 0.049), and the body weight of the VIAF group was extremely significantly higher than that of the NIAF group at 114 days (P = 0.008); The peak cumulative body weight of the NIAF cubs during lactation appeared around 130 days of age (n = 3); The mortality rate was 66.7% (n = 3) after about 150 days if the feed was continuously consumed nonvoluntarily. It was concluded that the milk secretion period of the mother is about 0–5 months after giving birth; the weaning period of the cubs should be 4–5 months after birth. If the cubs don't follow the mother to eat artificial feed for 3 months after birth, it can start be induced with artificial diet which adds termites, and the time point cannot be later than 130 days, otherwise it is not conducive to the survival of the cubs; When sexually mature, the body length and body weight of female cubs account for about 84% and 60% of the adult, respectively; the body maturity and body weight of female cubs tend to be stable about 15.3 months and 16.4 months, respectively. Finally, a special needle-shaped nipples and nursing patterns of female Sunda pangolins were also recorded in this study. These findings play an important role in guiding the nursing of captive Sunda pangolin cubs and other pangolin cubs. It is expected to improve the survival rate of the cubs by exploring the appropriate weaning time and the rules of weight growth. By scientifically planning the reproductive cycle of the female Sunda pangolins, our goal is to expand the population size and eventually release to the wild, meanwhile improving knowledge of this critically endangered species.

Administration of China (Reference number: 2019072), Guangxi Forestry Bureau (Reference numbers: GL2018kt-17 and GL2020kt-25). The funders had no role in study design data collection and analysis, decision to publish, or preparation of the manuscript.

**Competing interests:** The authors have declared that no competing interests exist.

## Introduction

The Sunda pangolin (*Manis javanica*) is one of the eight known pangolins of the Order Pholidota, Family Manidae [1]. Sunda pangolins were mainly distributed in the Indochina Peninsula, Malay Peninsula, Indonesia and southern Yunnan of China [2, 3]. In recent decades, the number of wild Sunda pangolins has sharply declined due to overexploitation and destruction of natural habitats [3]. Based on this situation, in 2014 the International Union for Conservation of Nature (IUCN) upgraded the Sunda pangolin from Endangered to Critically Endangered on the Red List of Threatened Species [4]. In October 2016, the contracting parties passed a resolution to upgrade all the eight pangolin species from Appendix II to Appendix I at the 17th Conference of the Parties to the Convention on International Trade in Endangered Species of Wild Fauna and Flora (CITES) [5]. As a result, international commercial trade of pangolins has been completely banned since January 2, 2017 [5].

Captive breeding of endangered species is one of the important approaches to prevent the extinction. It is extremely difficult to collect data on Sunda pangolins in the wild because of their scarcity and nocturnal habits, and it is also difficult to breed them in captivity [6, 7]. In October 2021, we published a paper on the successful breeding of Malayan pangolin for three generations, prior to this, there were no reports of systematic breeding of Malayan pangolins in the second generation [8]. As a result, the biological and ecological data of this species, especially the reproduction data, are extremely limited [9, 10], data on growth and development, as well as nursing behavior, are also very rare. More information is still required to reveal the natural attributes of this species. Weaning time has a great influence on the growth and mortality of mammals [11]. Studies have shown that early weaning has a negative impact on growth. Therefore, timely weaning can effectively improve the survival rate and yield of animals [12]. In addition, weaning time can affect the diversity and composition of fecal microbiota. Massacci *et al.* (2020) said that later weaning increases the diversity of the gut microbiome, giving the pups a competitive advantage [13]. Moreover, weaning influences the immunity of the cubs. Studies have shown that weaning affects antibody responses to antigens given two weeks before weaning [14]. Therefore, determining the weaning period of Sunda pangolin is of great significance to the growth and development of young, improving survival rate and reducing mortality.

To figure out the key parameters of rearing cubs, in this study, the mass changes of captive Sunda pangolin cubs during lactation were statistically measured regularly, and a cubic equation model was constructed using SPSS25.0 to predict lactation and weaning periods. The female's growth rate was also monitored to reveal the biological characteristics of the species. This study aims to provide important reference for the protective captive breeding of critically endangered Sunda pangolin and the rescue of orphaned pangolins.

## Materials and methods

### Ethics statement

This study was approved by the Biology Ethics Committee of the Guangxi Forestry Research Institute (reference number: GFRIIACUC2016-001). The study was carried out in compliance with the guiding principles on the treatment of laboratory animals issued by the Ministry of Science and Technology and the Laboratory Animal Guideline for Ethical Review of Animal Welfare issued by the National Standard of the People's Republic of China and ARRIVE guidelines.

## Experimental animals

The experimental animals in this study were 13 descendants of Sunda pangolin born in the Rescue Center of Guangxi Forestry Research Institute from April 2016 to October 2019 (Table 1).

## Housing and nursing

Each female pangolin lived in a separate cage with the suckling offspring, and there was a surveillance camera above the activity area to monitor the nursing behavior. Both females and cubs were fed with the same artificial diet, which consisted of black ant powder, silkworm pupa powder, mealworm powder, soy protein powder, termite mound mud and a small amount of vitamin complex. The ingredients were mixed with water until a semi-fluid concentration is reached. Pangolins were fed 250–400 mL of food once a day between 17:00–20:00. Clean water (100–200 mL) was also provided with a separate bowl. Termites were added when the cubs refused to eat the artificial diet.

The captive conditions, methods, part of the pangolin's behavior and pregnancy period of the Sunda pangolins in the rescue center have been previously described [8, 15, 16].

## Data collection

The cubs were weighed for the first time within 1–15 days after birth, and then all cubs were weighed around the 10th, 20th, and 30th of each month. For individuals C4, C6, C7, C10, CC4, C15 and C16, we continued to record their mass and length after successful weaning until maturity.

## Estimated lactation period and weaning period

Individuals C4, C5, C7, C10, and C11 began to eat artificial diets autonomously at 86–108 days of age. We classified them as the VIAF group. C1, C2, and C8 had inflection points in weight gain at 133, 131, and 127 days after birth, respectively, due to their inability to take artificial food on their own, and C3 started eating at 127 days. We classified these four cubs as the NIAF group. If the cubs continue to rely exclusively on maternal milk, body weight will continue to drop after

**Table 1. Basic information of the 13 Sunda pangolin (*Manis javanica*) offspring and their mothers (recorded until October 31, 2019).**

| Offspring ID | Mother ID | Date mother received | Place of conception | Date of birth | Status | Survival time (days) |
|---|---|---|---|---|---|---|
| C1 | F8[1] | 2015.12.15 | Wild | 2016.04.29 | Died on 2016.09.30 | 154 |
| C2 | F12[1] | 2016.01.19 | Wild | 2016.05.01 | Alive | 1278 |
| C3 | F11 | 2016.01.19 | Wild | 2016.06.05 | Alive | 1243 |
| C4 | F3 | 2014.01.10 | Center | 2016.06.25 | Died on 2018.06.08 | 713 |
| C5 | F16[1] | 2016.04.16 | Wild | 2016.07.04 | Died on 2019.05.24 | 1054 |
| C6 | F15 | 2016.04.16 | Wild | 2016.08.22 | Alive | 1165 |
| C7 | F6 | 2015.09.04 | Center | 2016.10.18 | Died on 2018.07.07 | 627 |
| C8 | F8[2] | 2015.12.15 | Center | 2017.04.15 | Died on 2017.09.18 | 156 |
| C10 | F5 | 2015.09.04 | Center | 2017.05.27 | Alive | 887 |
| C11 | F16[2] | 2016.04.16 | Center | 2017.07.14 | Alive | 839 |
| C15 | F8[3] | 2015.12.15 | Center | 2018.03.31 | Alive | 579 |
| C16 | F12[2] | 2016.01.19 | Center | 2018.05.19 | Alive | 530 |
| CC4 | C3 | 2016.06.05 | Center | 2018.06.06 | Alive | 512 |

[1-3]represent the order in which the offspring from each female were born.

C represents first-generation offspring, CC represents second-generation offspring, and F represents wild female generation.

reaching the peak, indicating that maternal milk supply is insufficient to maintain the growth of cubs, or that lactation has stopped completely. Combined with the videos of nursing behaviors in the late lactation period, we were able to estimate the weaning and lactation periods.

## Signs of sexual and physical maturity

Since female Sunda pangolins do not have obvious estrus characteristics at sexual maturity [8], we facilitated their cohabitation with males at random times. The first conception of a particular female was regarded as sexual maturity for that female. Body was considered mature when body length (± 1 cm) and body weight (± 0.5 kg) gradually stabilize. We defined body length as the length from the tip of the nose to the tip of the tail when the pangolin stretches its body. We tracked and recorded the body length and body weight of C4, C6, C7, C10, CC4, C15, and C16 until sexual and physical maturity.

## Data analysis

Eleven models, including logistic, quadratic, and cubic equations, were constructed using SPSS 25.0 to fit changes in cubs' body weight. The best-fit model was selected to estimate the daily body weight of each cub and to estimate the point at which a significant difference between the VIAF group and the NIAF group began to appear.

The measured values used to estimate the mass of C1, C2 and C8 ranged from birth to the peak cumulative weight, and C3 ranged from birth to 127 days of age. The measured values of the VIAF group ranged from birth to 150 days of age. Models were fitted to the two groups as well as to each cub individually, and independent samples *t*-test was performed on the estimated daily mass from birth to 150 days of age.

## Results

### Mass changes and weaning of Sunda pangolin cubs

C1 showed peak cumulative mass on day 133 after birth (weight 1025 g), while C2 showed peak cumulative mass at 131 days of age (1210 g) (Fig 1 and S1 Table). Due to possible reasons

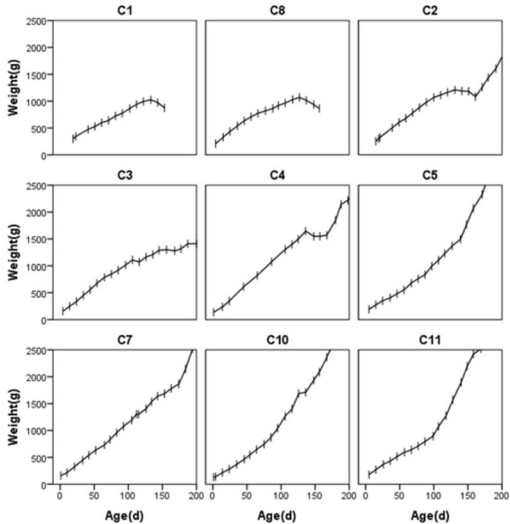

**Fig 1. Cumulative body mass growth curves for nine Sunda pangolin (*Manis javanica*) cubs from birth to 200 days of age.**

such as reduced milk and involuntary intake of artificial food, the mass of the two cubs dropped by 50 g and 20 g respectively 10 days after the peak. C1 died 21 days after the peak mass was reached (154 days of age, 875 g) (Fig 1). C2 began to eat a small amount of food at 150 days of age but continued to lose mass. By 161 days of age, C2' mass dropped to its lowest post-peak level of 1085 g but rebounded rapidly during the next ten days (Fig 1). There was no mass loss before C3 started to eat artificial food at the age of 127 days (Fig 1). C8 reached a peak body weight at 127 days of age (1070g) (Fig 1 and S1 Table). At the age of 140 days, we started coaxing the cubs to eat the artificial diet baited with termites, but it failed, and C8 died at 865g at 156 days after birth.

The peak mass of C1, C2, and C8 appeared at 133, 131, and 127 days of age, respectively. Afterwards, there was a brief drop in body mass, indicating that the female's milk supply was insufficient or even stopped secreting milk in the late lactation period. With the exception of C4 being forced to separate from its mother and losing weight, the VIAF group (including C3 who started to eat the artificial diet at 127 days of age) did not experience sustained weight loss. Therefore, we speculate that the theoretical weaning period of Sunda pangolin cubs is about 4–5 months after birth.

Monitoring video showed that the six-month-old male cub C5 was still suckling in the late lactation period (Fig 2A). Based on this, it is inferred that the lactation period of Sunda pangolins may last for 5–6 months.

### Estimated mass of Sunda pangolin cubs during lactation period

**Mass estimation for the NIAF group (C1, C2, C3, C8).** The results showed that the cubic equation model best fit the growth rates of the NIAF group, and its fit ($R^2$) ranks first among

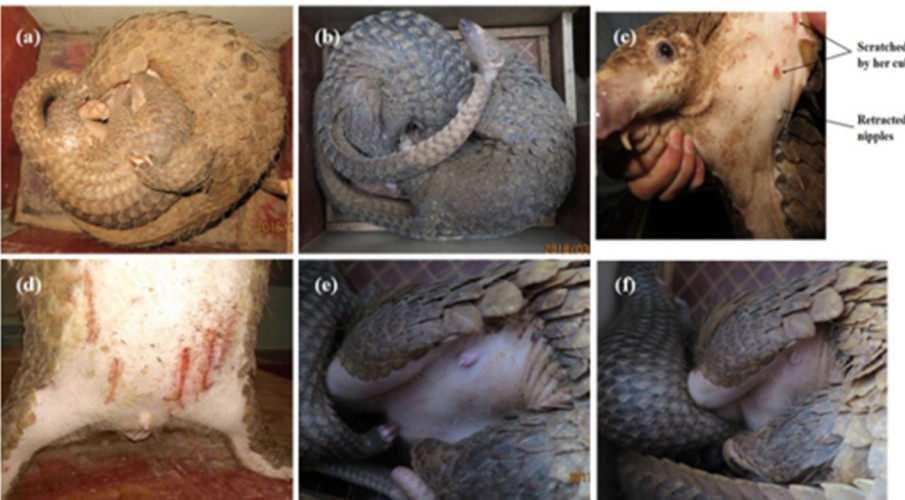

**Fig 2. (a)** The male Sunda pangolin (*Manis javanica*) cub C5 (mass 2640 g, body straight dorsal length 66 cm) was still suckling at 6 months of age. (Photo: Dingyu Yan, 28 December 2016). **(b)** At 10 months of age, female Sunda pangolin (*Manis javanica*) cub C10 (mass 4220 g, body straight dorsal length 80 cm) was still sharing her mother's denning box. (Photo: Dingyu Yan, 15 March 2018). **(c)** Sunda pangolin (*Manis javanica*) mother F3 was scratched by her 145-day-old male offspring while she was nursing him. (Photo: Dingyu Yan, 6 November 2016). **(d)** 7-month-old male Sunda pangolin (*Manis javanica*) cub C11 (mass 3330 g, body straight dorsal length 73 cm) was scratched by his mother while cohabiting (Photo: Dingyu Yan, 11 February 2018). **(e)** Needle-like single main duct nipple of a female Sunda pangolin (*Manis javanica*) that has been partially retracted just after lactation. (Photo: Dingyu Yan, 21 May 2017). **(f)** Needle-shaped nipples of a female Sunda pangolin (*Manis javanica*) retracted about 15 seconds after the cub stopped suckling. (Photo: Dingyu Yan, 21 May 2017).

the 11 models, with an $R^2$ of 0.952 (Fig 3A). The equation is $y = 136.47+9.79x-0.00463x^2-0.00009865x^3$, and the body mass fitting values of different ages are shown in S2 Table.

**Mass estimation for the VIAF group (C4, C5, C7, C10, C11).** Like the NIAF group, the cubic equation is the model with the best fit for VIAF group with an $R^2$ of 0.973 (Fig 3B). The equation is $y = 124.16+9.553x-0.039402x^2+0.00039x^3$, and the body mass fitting values of different ages are shown in S2 Table.

**Mass difference estimate between the two groups.** The fitted body mass curves and $R^2$ value of each cub are shown in Fig 4. The regression equation for the growth curve of each cub is shown in S3 Table. There was a significant difference in body mass between the two groups at 105 days of age (P = 0.049), while extremely significant differences began to appear at 114 days of age (P = 0.008).

## Estimating the mass and length of female Sunda pangolins at sexual and physical maturity

Female Sunda pangolins have an average body weight of 3255 g when sexually mature, and the average body straight dorsal length was 72 cm (n = 4) at 7–9 months. At 15.3 months old, the average body length of female stabilized at 86cm (n = 4), indicating physical maturity. At 16.4 months old, the average body weight stabilized at 5385 g (n = 7), indicating physical maturity (S4 Table). One of the growth information was shown in Fig 5. The body weight at sexual maturity accounts for about 60% of the body weight physical maturity, and the body length at sexual maturity accounts for about 84% of the body length at physical maturity.

## Characteristic of nursing behavior

Almost all the nursing behaviors were recorded by cameras in the activity area. Through monitoring, we found some nursing behaviors:

**Their mother would not expel or harm the cubs if they had not been separated since birth.** Although two dens were provided in each enclosure, mothers and cubs usually chose to live together. Even if cubs such as C3 and C4 gradually grew sharp claws to scratch their

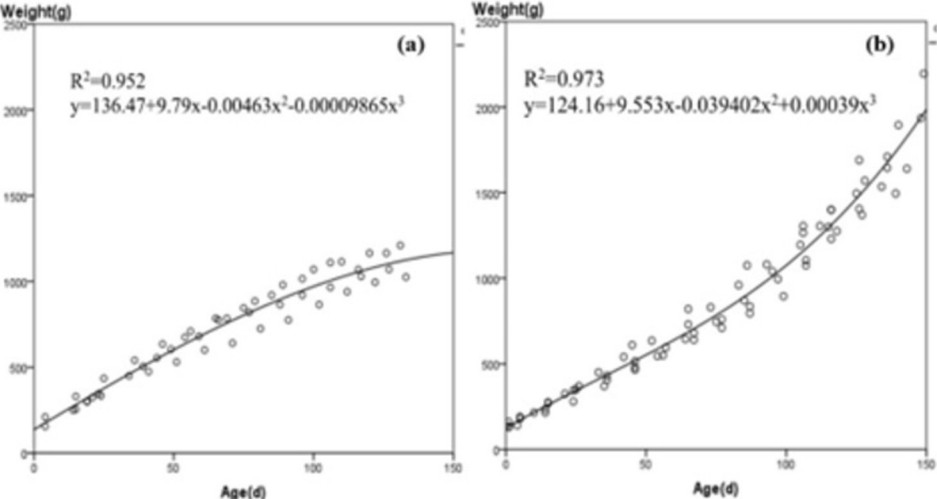

**Fig 3. (a)** Body mass as a function of age for four Sunda pangolin (*Manis javanica*) cubs under a natural feeding regime. **(b)** Body mass as a function of age for five Sunda pangolin (*Manis javanica*) cubs fed an artificial diet.

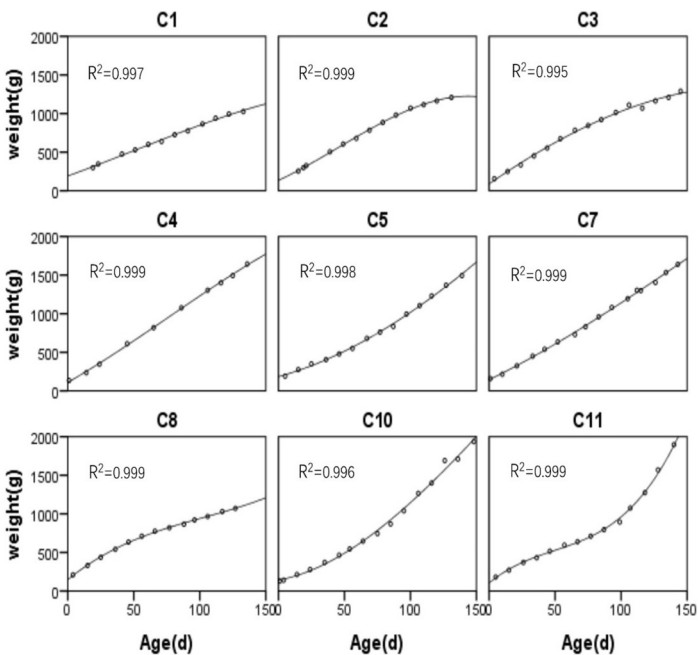

**Fig 4. Fitted body mass growth curves for each of the nine Sunda pangolin (*Manis javanica*) cubs.**

mothers, they did not get hurt. Male cub C5 is still suckling at nearly 6 months old, and female cub C10 is still sleeping in the same den with its mother at 10 months old (Fig 2A and 2B).

**In some cases, the mother may also harm the suckling cub.** 145-day-old C4 was forced to separate from its mother (F3) for two days because it scratched the skin around the mother's teat (Fig 2C). However, as C4 refused to eat when it was removed, we returned him to its

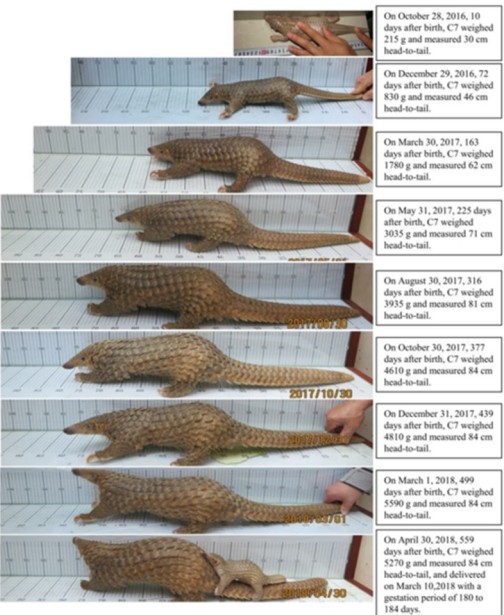

**Fig 5. The growth process of the first-generation offspring C7. (Photos: Dingyu Yan).**

mother. The next day, we found scratches on C4's abdomen from F3. Another example is that female F16 left the den to mate with a male, then returned to the den after a while and scratched her cub C11 (Fig 2D).

**There are no obvious signs of weaning between mother and cubs.**   145-day-old C4 had developed hard, sharp nails and was eager to scratch its mother's breasts when its appetite was not satisfied, then we separated C4 from its mother to protect the mother, which may be due to decreased or stopped milk secretion. The other cubs were separated from their mothers 180–363 days after birth. There were no obvious signs of weaning between mother and cubs, unless successfully weaned cubs were no longer dependent on mother's milk and could eat artificial food on their own. For example, the mother does not drive or harm the cub, and the cub usually lies on the mother's back or tail when entering or exiting the den.

**Special needle-shaped nipples.**   Female Sunda pangolins' nipples are needle-like and normally retracted, extending only when sucking or releasing by itself (Fig 2C, 2E and 2F). There were no significant changes in breast size or shape before mating, after conception or during lactation.

**An interesting feeding.**   A mother pangolin uses her tongue deliver artificial diet under the scales for its cub to lick (S1 File). As a result, the cub gradually accepted and was able to eat artificial diet on its own.

## Discussion

The estimated mass of the VIAF group and the NIAF group began to show a significant difference at 105 days, and an extremely significant difference appeared at 114 days. The peak mass of the three cubs in the NIAF group appeared at about 130 days of age, and two of these cubs died within 160 days after birth due to their inability to switch from mother's milk to artificial diet. Based on these results, the female's milk production appears to have dropped to a level that cannot sustain the growth of cubs by about 130 days postpartum. We believe that milk production and energy and nutrients intake may be reduced at this stage, which is not sufficient to support the survival of the cubs. Therefore, after reaching the peak weight, the weight will gradually decrease, until the body condition becomes weak, sick, and eventually leads to death. After a mother breastfed her pups for an extended period (late lactation period), it is difficult for milk to meet the nutritional needs of the cubs, and artificial feed needs to be supplemented. A study in rabbits showed that a group of pups that were exclusively breastfed developed iron-deficiency anemia, whereas a group of pups that ate an iron-supplemented diet had lower rates of iron-deficiency anemia [17]. Longer lactation periods also increase disease susceptibility, as they show lower numbers of total lymphocytes at the end of lactation [18]. In addition, a longer lactation period consumes maternal energy for a long time, which is also one of the reasons for reducing maternal immunity [19, 20]. Therefore, we recommend that cubs start be induced with artificial diet at 90 days old and transition to artificial diet before 130 days old, preferably not later than 130 days old.

In this study, the mothers were only fed an artificial feed. However, Sunda pangolins naturally eat termites and small amounts of other species of ants. This feeding habit is engraved in the genetic memory of the young. Furthermore, artificial food is completely different from termites in terms of smell and taste, and most of the cubs cannot naturally follow their mothers to learn how to eat the artificial food, which may have affected the weaning time of the cubs.

Weaning is an important part of the pangolin's life cycle and is related to survival rate. Termites are key to helping cubs transition to artificial formula–we have successfully weaned multiple cubs including C2 and C3 by using termites to coaxing cubs into eating an artificial diet. For cubs, creep feeding during pre-weaning lactation is a common husbandry practice because

it increases weaning weight of cubs and leads to a smooth transition period from milk to the artificial feed for cubs [21, 22]. Previous studies demonstrated that creep feed intake has a positive effect on post-weaning feed intake [21, 23]. At the same time, this promotes gut development and therefore helps them cope with dietary changes more easily after weaning [24]. But at present, the feeding effect of artificial feed induction is not ideal. The feed we give to captive pangolins consists of about 30–40% black ants. The synthetic feed mixed with silkworm chrysalis, and others are different in smell and teste from natural edible termites. In fact, black ants may not be the main natural food for *M. javanica*, as most of the wild pangolins we just rescued ate termites instead of black ants during induction feeding.

Meanwhile, the cubs' low acceptance of artificial feed made us to suspect that captive pangolins are weaning longer than wild pangolins. We speculate that wild Sunda pangolin cubs may start foraging earlier than those in captivity, so the mass of wild cubs of the same age during the mid-lactation period may be heavier than those in captivity. However, cubs raised in captivity during late lactation tend to gain mass faster in an environment with adequate food. No article has reported biological information about the growth of Sunda pangolins in the wild. Sun *et al*. (2018) observed that the overall growth rate of wild Chinese pangolin (*Manis pentadactyla*) cubs during lactation was 1.2 cm/week [25], which was faster than the growth rate of captive individuals (0.7 cm/week); [26].

Through behavioral observations, we observed a peculiar phenomenon of longer than normal breastfeeding. Monitoring video showed that the six-month-old male cub C5 was still suckling in the late lactation period. Research suggests that late lactation is thought to be more important for maintaining the mother-cub relationship than the true nutritional benefit to the pups [27, 28]. In other words, it is possible that the psychological effect of breastfeeding at this stage may be greater than the physiological effect.

We constructed the growth patterns of nine cubs based on recorded pup weight data, including the individual growth patterns of the nine pups, VIAF group and NIAF group. The cubic equation fitting model can best reflect the growth process of these individuals or groups. The fitting degrees of the nine individuals' models are all above 0.995, and the fitting degrees of the 2 group models are all above 0.95. The fitting degree is above 0.95, indicating that the estimated value can better reflect the actual value, especially the individual model, with a fitting degree of 0.995. There was a significant difference in body weight between the VIAF group and the NIAF group at 105 days, and an extremely significant difference in body weight began to appear between the two groups at 114 days. These data provide a theoretical reference for us to determine the food induction time of young pangolin.

Payne and Francis (1998) speculate that the lactation period of Sunda pangolins was about 3 months [29], while Lim and Ng (2008) used radio tracking technology to study the mother-child relationship of wild Sunda pangolins and estimated the lactation period is about 3–4 months [30]. Zhang *et al*. (2015) estimated the weaning period to be about 4 months [31].

In these articles, the authors estimated the weaning period based on the behavior between mother and cub. In this study, weaning period was estimated to be 4–5 months according to the age of growth inflection point and death time of NIAF individuals, as well as the time of significant weight difference between NIAF group and VIAF group. According to our observation, we could not definitively judge the weaning period from the behavior of mother and cub. Our results rely primarily on detailed data on cub weight growth and the establishment of growth model projections, an innovative approach that theoretically explains the observations.

We observed that mothers do not expel or harm the cubs after weaning if they are not separated after birth. However, in some cases, the mother may also harm the cub. When we tried to separate two cubs from their mother for a few days and return them, we found that the mother was harming her cubs. This may be due to the cub is looking for the mother's milk, but

the mother doesn't seem to recognize her baby and fights back, seeing it as an aggressive act. Challender *et al.* (2012) observed "wrestling" (repelling) behavior between a female Sunda pangolin and her 4-month-old cub [32]. The explanation for this is that in the late lactation, the young are no longer completely dependent on their mother's milk, and they forage separately from each other. After reunion, the mother was disturbed by the cub's attempts to get close to her, which leds to "wrestling" (repelling) behavior that harmed the cub. Common domestic animals can be weaned without conflict between the mother and the cubs. In most cases, the pups are forced to be weaned in isolation before natural weaning, that is, before the end of the mother's lactation period. However, there have also been reports of female calves being prevented from sucking by their mothers during natural weaning, but not harmful behavior [33], while in wild animals, such as baboons, weaning conflicts in baboons can last for weeks or months, including daily competitive interactions and loud crying from babies [34]. The reduction in milk intake appears to be a matter of reduced milk availability rather than a reduction in milk production (i.e., the reduction in milk intake is more of a behavioral process than a physiological process). A reduction in the mother's tendency to feed her pups triggers the weaning process, which will be driven by the mother's perception of the pup's physical, physiological, and behavioral characteristics [35] rather than the length of time after delivery [36].

In addition, we observed that mother would feed artificial diets to her cubs (Supplementary video in S1 File). This behavior was only observed between a mother and a cub. Are there any mothers in the wild who use its long tongues to stick termites or ants in gaps to feed its cubs? Peer needs to observe and verify this interesting behavior.

Furthermore, the breast of Sunda Pangolins is very different from most other mammals we commonly see. We suspect that needle-shaped nipples are more susceptible to be stabbed by acanthids and bitten by termites, so that placing the breast under the arm and retracting it is a highly effective protection strategy. However, contracted nipples make it more difficult for babies to suck. Therefore, we suspect that when cubs are suckling, the mother's nipples are actively stretching when they are sucked or touched around the breast.

This research aimed to determine the growth characteristics of Sunda pangolin cubs during lactation. The lactation and weaning periods of Sunda pangolins were estimated according to the time taken to reach peak mass, the time at which cubs began to eat artificial diet on their own and the age at which cubs that did not transition to the artificial diet died. Few similar analyses have been done to determine weaning, whether domesticated domestic animals or captive wild mammals, whose offspring eat a diet close to their natural diet. So, the offspring of these animals are fed artificial food before the end of their mother's lactation and their pups are weaned smoothly. Because pangolin's natural diet is so special, our study is unique and meaningful. This study also shows the nursing behavior of Sunda pangolin, revealing the indicators of sexual maturity and physical maturity. The artificial formula used in this study is quite different from the natural food of Sunda pangolins. Therefore, the conclusions drawn in this article may only apply to the physiological information of Sunda pangolins in captivity. Although the sample size of this study is limited, it is preliminary. There is currently a lack of data on wild Sunda pangolins which precludes any comparisons being drawn, but the results of this study can still be used as a reference for artificial rescue and domestication of pangolins and provide valuable basic data for understanding this mysterious animal.

## Supporting information

**S1 Table. Growth information of nine Sunda pangolin (*Manis javanica*) cubs.**
(XLSX)

**S2 Table. Fitted values of the feeding (n = 4) and non-feeding (n = 5) groups of Sunda pangolins (*Manis javanica*) at different ages.**
(XLSX)

**S3 Table. The regression equation of the growth curve for each cub.**
(XLSX)

**S4 Table. Details regarding the length and mass of female Sunda pangolins (*Manis javanica*) at maturity.**
(XLSX)

**S1 File. Supplementary video.**
(MP4)

## Acknowledgments

We thank Prof. Darren Pietersen of the University of Pretoria for suggesting revisions to this article. We would also like to thank the Guangxi Terrestrial Wildlife Rescue Research and Epidemic Disease Monitoring Centre for providing animals for this study.

## Author Contributions

**Data curation:** Dingyu Yan, Xiangyan Zeng, Baocai Li, Jinyan Chen.

**Formal analysis:** Xiaobing Guo.

**Funding acquisition:** Dingyu Yan.

**Project administration:** Dingyu Yan.

**Software:** Dingyu Yan, Xiangyan Zeng.

**Writing – original draft:** Dingyu Yan, Miaomiao Jia, Xiaobing Guo.

**Writing – review & editing:** Tengcheng Que, Li Tao, Mingzhe Li, Shanghua Xu, Yan Hua, Shibao Wu, Peng Zeng, Shousheng Li, Yongjie Wei.

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
