## [Decision Letter · Decision Letter 0]

13 May 2022

PONE-D-22-10275Weaning period and growth patterns of captive Sunda pangolin (Manis javanica) cubsPLOS ONE

Dear Dr. Yan,

Thank you for submitting your manuscript to PLOS ONE. After careful consideration, we feel that it has merit but does not fully meet PLOS ONE’s publication criteria as it currently stands. Therefore, we invite you to submit a revised version of the manuscript that addresses the points raised during the review process.

We look forward to receiving your revised manuscript.

Kind regards,

Bi-Song Yue, Ph.D

Academic Editor

PLOS ONE

Journal Requirements:

"This study was funded by the Nature Science Foundation of Guangxi (2018GXNSFAA294066), the State Forestry Administration of China (Reference number: 2019072), Guangxi Forestry Bureau (Reference numbers: GL2018kt-17 and GL2020kt-25)."

"This study was funded by the Nature Science Foundation of Guangxi (2018GXNSFAA294066), the State Forestry Administration of China (Reference number: 2019072), Guangxi Forestry Bureau (Reference numbers: GL2018kt-17 and GL2020kt-25). "

"This study was funded by the Nature Science Foundation of Guangxi (2018GXNSFAA294066), the State Forestry Administration of China (Reference number: 2019072), Guangxi Forestry Bureau (Reference numbers: GL2018kt-17 and GL2020kt-25)."

Reviewers' comments:

Reviewer's Responses to Questions

**Comments to the Author**

1. Is the manuscript technically sound, and do the data support the conclusions?

Reviewer #1: No

Reviewer #2: Yes

2. Has the statistical analysis been performed appropriately and rigorously? 

Reviewer #1: No

Reviewer #2: Yes

3. Have the authors made all data underlying the findings in their manuscript fully available?

Reviewer #1: Yes

Reviewer #2: Yes

4. Is the manuscript presented in an intelligible fashion and written in standard English?

Reviewer #1: No

Reviewer #2: Yes

5. Review Comments to the Author

Reviewer #1: The submitted manuscript focuses on establishing the weaning age and estimating the growth patterns of captive Sunda pangolin. The work presented in this study should be treated as very preliminary results.

Major issues

1. The aim of the study is not clearly stated in the introduction.

2. Abstract should be rewritten as it does not present clearly the study and its results.

3. The authors estimate growth patterns of several animals, but they never discuss it in the discussion. What growth patters gave your study? What did you learn from this? How can it help captive Sunda pangolin in the future? Was the same analysis used in different studies on captive animals? How did it help those species?

Also it should be clearly noted that the number of animals used in this study should does not allow to make strong conclusions and should be treated as a very preliminary study.

4. Most of discussion should be placed in results in section on

"video observations", which is completely missing from the "Results". Style of writing of the discussion is not very scientific. There is very little references used, which is understandable when knowledge on the topic is scare, however, there is no comparison to similar studies on other captive species. This would give the reader more perspective on where this research could lead.

Thus, I highly recommend to the Authors to place this study in the more global perspective and compare it with similar works in different captive species. Without is this work is not even the preliminary results, but merely a guess based on very imited data.

Minor issues

Table S2 - "fitted values" of what?

Wrong order of Figures in the text.

LINE 179-187 - please keep the order models as the animlas from in table or clearly divide them in two groups of animals.

Figure S1 - should be in main text

Reviewer #2: The research is well designed and implemented. Due to the lack of information about wild Sunda pangolins, the information obtained from this article seems to provide valuable basic data for domestication and rescuing orphaned pangolins.

6. PLOS authors have the option to publish the peer review history of their article (what does this mean?). If published, this will include your full peer review and any attached files.

Reviewer #1: No

Reviewer #2: No

---

## [Author Response · Author response to Decision Letter 0]

2 Jul 2022

R: Requirement, Q: Question, A: Answer.

Journal Requirements:

R1. Please ensure that your manuscript meets PLOS ONE's style requirements, including those for file naming. The PLOS ONE style templates can be found at https://journals.plos.org/plosone/s/file?id=wjVg/PLOSOne_formatting_sample_main_body.pdf and https://journals.plos.org/plosone/s/file?id=ba62/PLOSOne_formatting_sample_title_authors_affiliations.pdf

A1: We have revised the manuscript according to PLOS ONE’s style requirements.

R2. Thank you for stating the following financial disclosure:

"This study was funded by the Nature Science Foundation of Guangxi (2018GXNSFAA294066), the State Forestry Administration of China (Reference number: 2019072), Guangxi Forestry Bureau (Reference numbers: GL2018kt-17 and GL2020kt-25)."

A2: We have stated “The funders had no role in study design data collection and analysis, decision to publish, or preparation of the manuscript” in our cover letter.

R3. Thank you for stating the following in the Acknowledgments Section of your manuscript:

"This study was funded by the Nature Science Foundation of Guangxi (2018GXNSFAA294066), the State Forestry Administration of China (Reference number: 2019072), Guangxi Forestry Bureau (Reference numbers: GL2018kt-17 and GL2020kt-25). "

"This study was funded by the Nature Science Foundation of Guangxi (2018GXNSFAA294066), the State Forestry Administration of China (Reference number: 2019072), Guangxi Forestry Bureau (Reference numbers: GL2018kt-17 and GL2020kt-25)."

A3: The funding-related text has been removed from the manuscript. In addition, the funding statement has not been updated.

Q4. Please include your full ethics statement in the ‘Methods’ section of your manuscript file. In your statement, please include the full name of the IRB or ethics committee who approved or waived your study, as well as whether or not you obtained informed written or verbal consent. If consent was waived for your study, please include this information in your statement as well.

A4: The ethics statement has been included in the ‘Materials and methods’ section of the revised manuscript.

Reviewers' comments:

Reviewer's Responses to Questions

Comments to the Author

1. Is the manuscript technically sound, and do the data support the conclusions?

Reviewer #1: No

Reviewer #2: Yes

2. Has the statistical analysis been performed appropriately and rigorously?

Reviewer #1: No

Reviewer #2: Yes

3. Have the authors made all data underlying the findings in their manuscript fully available?

Reviewer #1: Yes

Reviewer #2: Yes

4. Is the manuscript presented in an intelligible fashion and written in standard English?

Reviewer #1: No

Reviewer #2: Yes

5. Review Comments to the Author

Reviewer #1: The submitted manuscript focuses on establishing the weaning age and estimating the growth patterns of captive Sunda pangolin. The work presented in this study should be treated as very preliminary results.

Major issues

Q1. The aim of the study is not clearly stated in the introduction.

A1: The introduction has been revised as required, and the aim and significance are summarized at the end. Please refer to line 65-102 of the revised manuscript. 

Q2. Abstract should be rewritten as it does not present clearly the study and its results.

A2: Abstract has been rewritten as required, please refer to line 34-62 of the revised manuscript.

Q3. The authors estimate growth patterns of several animals, but they never discuss it in the discussion. What growth patters gave your study? What did you learn from this? How can it help captive Sunda pangolin in the future? Was the same analysis used in different studies on captive animals? How did it help those species? 

Also it should be clearly noted that the number of animals used in this study should does not allow to make strong conclusions and should be treated as a very preliminary study.

A3: Growth patterns of nonvoluntarily ingested artificial feed (NIAF) and voluntarily ingested artificial feed (VIAF) groups and individual growth patterns of all 9 pups. The cubic equation fitting model best reflects the growth process of these individuals or groups. The fitting degree of the 9 individual fitting models was above 0.995, and the fitting degree of the 2 group models was above 0.95.

What did you learn from this?

The captive breeding of pangolins is a worldwide problem, and there is no reference to pangolin breeding before this. We are the first to report in detail the growth pattern of pangolin species (one of 8) cubs, and determine the weaning period and lactation period by studying the growth pattern, formulate a parenting plan, and decide when to induce the pups to eat artificial feed. Improving the survival rate of pangolin pups after weaning.

How can it help captive Sunda pangolin in the future?

Three pangolin species, including the Sunda pangolin, are critically endangered species, and ex situ conservation is also an important option while strengthening wild protection. This study can provide valuable references for the conservation of pangolins and the rescue of orphaned pangolins.

Was the same analysis used in different studies on captive animals?

There are few similar analyses to determine the weaning period, as both domesticated domestic animals and captive wild mammals are given feeds that are close to their natural diets in the management of their offspring. Therefore, the offspring of these animals start to eat artificial feed before the end of lactating, and the cubs can be weaned smoothly. Because the natural food of pangolins is too special, the feed we give to captive pangolins is a synthetic feed composed of about 30-40% black ants, plus silkworm pupae, etc. The smell and taste of their natural food termites are too different. In fact, black ants may not be the main natural food of Malay pangolins, because most of the wild pangolins that we just received rescued did not eat black ants, but termites. The cub's genetic memory can only eat termites, and although the mother eats artificial food, it cannot follow the food.

How did it help those species? 

This research set out to determine the growth characteristics of Sunda pangolin cubs during lactation. The information obtained from this article can provide valuable basic data for domestication and rescuing orphaned pangolins.

Also it should be clearly noted that the number of animals used in this study should does not allow to make strong conclusions and should be treated as a very preliminary study.

We agree with the reviewers.

Q4. Most of discussion should be placed in results in section on "video observations", which is completely missing from the "Results". Style of writing of the discussion is not very scientific. There is very little references used, which is understandable when knowledge on the topic is scare, however, there is no comparison to similar studies on other captive species. This would give the reader more perspective on where this research could lead.

Thus, I highly recommend to the Authors to place this study in the more global perspective and compare it with similar works in different captive species. Without is this work is not even the preliminary results, but merely a guess based on very imited data.

A4: We have adjusted some content in Results and Discussion. And we have added references and their content, and discussed.

Q5: Table S2 - "fitted values" of what?

A5: The fitted value is the point estimate of the mean response to a given predictive variable value

Q6: Wrong order of Figures in the text.

A6: The order of Figures has been corrected.

Q7: LINE 179-187 - please keep the order models as the animals from in table or clearly divide them in two groups of animals.

A7: We have changed the way data and results are presented (S3).

Q8: Figure S1 - should be in main text

A8: We have made changes based on the reviewer's comments.

Reviewer #2: The research is well designed and implemented. Due to the lack of information about wild Sunda pangolins, the information obtained from this article seems to provide valuable basic data for domestication and rescuing orphaned pangolins.

---

## [Editor Report · Decision Letter 1]

12 Jul 2022

Weaning period and growth patterns of captive Sunda pangolin (Manis javanica) cubs

PONE-D-22-10275R1

Dear Dr. Yan,

We’re pleased to inform you that your manuscript has been judged scientifically suitable for publication and will be formally accepted for publication once it meets all outstanding technical requirements.

Kind regards,

Bi-Song Yue, Ph.D

Academic Editor

PLOS ONE

---

## [Editor Report · Acceptance letter]

23 Aug 2022

PONE-D-22-10275R1 

Weaning period and growth patterns of captive Sunda pangolin (*Manis javanica*) cubs 

Dear Dr. Yan:

I'm pleased to inform you that your manuscript has been deemed suitable for publication in PLOS ONE. Congratulations! Your manuscript is now with our production department. 

Kind regards, 

on behalf of

Dr. Bi-Song Yue 

Academic Editor

PLOS ONE